# Persistent Postural-Perceptual Dizziness (PPPD) from Brain Imaging to Behaviour and Perception

**DOI:** 10.3390/brainsci12060753

**Published:** 2022-06-08

**Authors:** Patricia Castro, Matthew J. Bancroft, Qadeer Arshad, Diego Kaski

**Affiliations:** 1Neuro-Otology Department, University College London Hospitals, London WC1E 6DG, UK; p.castro-abarca16@imperial.ac.uk; 2Department of Brain Sciences, Imperial College London, London W6 8RF, UK; 3Departamento de Fonoaudiología, Facultad de Medicina, Universidad de Chile, Santiago 8380453, Chile; 4Centre for Vestibular and Behavioural Neuroscience, Department of Clinical and Movement Neurosciences, Institute of Neurology, University College London, London WC1N 3BG, UK; matthew.bancroft.13@ucl.ac.uk; 5Department of Neuroscience, Psychology and Behaviour, University of Leicester, Leicester LE1 7RH, UK; qa15@leicester.ac.uk

**Keywords:** persistent postural perceptual dizziness, PPPD, functional imaging, vestibular net-works, perception, behaviour, postural control

## Abstract

Persistent postural-perceptual dizziness (PPPD) is a common cause of chronic dizziness associated with significant morbidity, and perhaps constitutes the commonest cause of chronic dizziness across outpatient neurology settings. Patients present with altered perception of balance control, resulting in measurable changes in balance function, such as stiffening of postural muscles and increased body sway. Observed risk factors include pre-morbid anxiety and neuroticism and increased visual dependence. Following a balance-perturbing insult (such as vestibular dysfunction), patients with PPPD adopt adaptive strategies that become chronically maladaptive and impair longer-term postural behaviour. In this article, we explore the relationship between behavioural postural changes, perceptual abnormalities, and imaging correlates of such dysfunction. We argue that understanding the pathophysiological mechanisms of PPPD necessitates an integrated methodological approach that is able to concurrently measure behaviour, perception, and cortical and subcortical brain function.

## 1. Introduction

Persistent postural-perceptual dizziness (PPPD) is one of the most common causes of chronic dizziness, affecting 4% of individuals registered with general practitioners in the United Kingdom [1]. However, our clinical experience, supported by empirical data [2], is that self-reported symptoms of PPPD are underreported.

In 2017, a consensus committee of the Bárány society proposed diagnostic criteria for PPPD that included a sensation of unsteadiness and non-spinning vertigo, exacerbated by upright posture, active or passive motion, and exposure to moving visual stimuli or complex visual patterns [3]. It is associated with significant morbidity, owing both to its chronicity but also the nature of the perceptual disturbance that impacts upon physical and cognitive function [4]. The clinical features that support its diagnosis (Figure 1) include a preceding balance insult (usually, but not exclusively, vestibular) followed by chronic dizziness often with accompanying psychological symptoms, typically displayed during a normal neurological examination [3].

However, the clinical phenotype can be diverse. Preceding trait anxiety and psychological stress are significant risk factors for the development of PPPD [5,6], with other personality traits, such as neuroticism, also influencing the development of chronic dizziness symptoms, leading to development of a hypervigilant state and introspective self-monitoring [7]. Negative illness perception may also be a factor in the development of PPPD, and such factors may contribute to its pathophysiology [7]. Other non-psychological variables are also involved in PPPD, such as lower sensory thresholds with increased sensitivity to stimuli across the range of sensory modalities, which is not fully explained by anxiety [8], with lack of habituation [9] and increased visual reliance during sensory integration [10]. 

Theoretical considerations to explain the pathophysiology of PPPD are emerging based on our understanding of the precursors to PPPD, namely chronic subjective dizziness, phobic postural vertigo, and visual vertigo [4,11,12]. On the one hand, increased self-motion perception, and thus, heightened instability, may result from a mismatch between expected (efferent) and actual (afferent) motion signals [13,14,15]. This may relate to a reduced tolerance for errors between estimates and sensory inputs [16]. Indeed, it has been suggested that the brain uses generative models to actively construct explanations to infer the causes of sensory inputs, such that the brain’s internal model of the external world is optimised to provide predictable sensory inputs [17,18]. The prediction error is therefore the difference between the input observed and that predicted by the generative model and inferred causes. The concept of predictive coding describes the brain’s attempt to minimise surprise and resolve uncertainty about sensory information, reducing the mismatch between unpredicted (“salient”) sensations and those predicted under the generative model [17]. Only unpredicted inputs must then be further transmitted and analysed. The abnormal processing of seemingly “unpredicted” signals may account for abnormal prediction error and the increased perception of sway in patients with PPPD. Within this framework of predictive coding, the central processing of incoming sensory information is biased by a mismatch resulting from incorrect internal expectations, leading to the perception of impaired posture or balance. On the other hand, patients with PPPD exhibit heightened awareness or attention to moving or complex visual stimuli [10,19,20,21,22], but also increased sensitivity to self and object motion [23]. 

Understanding the underlying pathophysiology of PPPD symptoms has more recently focused on imaging modalities. Here, we review the clinical features and brain imaging findings in PPPD and discuss the importance of behavioural and perceptual experimental data in better understanding the pathophysiology of this disorder. 

## 2. Clinical Features 

Patients with PPPD present with two key fluctuating or continuous symptoms: (a) a dizzy, not-truly-vertiginous sensation, with patients reporting that their head is swimming, and/or (b) unsteadiness, such that patients report swaying, rocking, or “jelly legs” [4]. Symptoms can be exacerbated in visually complex environments and during upright posture and head movements. Patients often report feelings of disconnectedness from the self and their surroundings (mild depersonalisation and derealisation). Such low-level but mostly persistent dissociative symptoms are very commonly reported by patients with PPPD, but also experienced by healthy individuals who are exposed to a strong vestibular stimulus [24,25,26], suggesting such cognitive perceptions are indeed engendered by vestibular symptoms. The character of dissociative symptoms includes a feeling of brain fog or non-specific sensations of disorientation, together with a range of more specific cognitive symptoms that include short-term memory loss, difficulty concentrating, and impairments in multi-tasking (Figure 1) [27]. 

Typically, PPPD is triggered by an acute disruption to normal balance function, whether triggered by a vestibular or non-vestibular stimulus. In the former, the development of PPPD is not, however, attributable to the degree of inner ear (otological) dysfunction or failed ear recovery [3]. Rather, psychological risk factors and shifts in psychophysical functioning appear to strongly predict which patients will go on to develop PPPD following a triggering event [7]. Many of the clinical features of PPPD are physiological consequences of an acute balance dysfunction: shifting balance and motion strategies to visually and posturally based inputs is adaptive in vestibular failure [28]. A lack of re-adaptation following resolution of the vestibular insult (or in persistent vestibular dysfunction) can result in the chronic adoption of poor postural and gait strategies, such as a stiffened walk and shorter strides, and increased visual dependence [29]—attempts to adapt to a perceived postural threat that no longer exists, which thus become maladaptive. (Figure 1).

## 3. Brain Imaging

A systematic literature review was completed to identify original, English-language research articles using neuroimaging to investigate patients with PPPD as defined in the ICD-11 criteria (ICD-11, 2005), published between 1 January 2015 and 30 September 2020. The information sources used in this review were Medline/Ovid, Embase/Ovid and PubMed. The search strings included the following key terms “Vestibular Diseases” OR “Dizziness” OR “persistent postural perceptual dizziness”; OR “persistent perceptual postural dizziness”; OR (“persistent postural” and “perceptual dizziness”) AND pathophysiology OR physiopathology OR pathophysiological mechanism; OR pathogenesis OR pathological OR “psychophysics”. The search criteria returned 3376 articles, reduced to 2443 after the removal of duplicates. After screening the title and abstract, a further 2405 were excluded as irrelevant. The remaining 38 full-text articles were assessed for eligibility and 30 were excluded because: they exclusively related to PPPD (*n* = 1), were review articles (*n* = 13), not related to the pathophysiology of PPPD (*n* = 9) and did not include neuroimaging (*n* = 7) (Figure 2). The remaining eight articles were evaluated for the risk of bias, analysed and the appropriate data were extracted. For a comprehensive imaging review, see [30].

There can be little doubt that PPPD is a neuropsychological disorder and on this basis, there has been an assumption that patients may manifest alterations in brain function, or even structure. Are, however, the brains of patients with PPPD—or, for that matter, anxiety or neuroticism—truly different from those without, or are any neural changes perhaps transient in such individuals, mirroring their neuropsychological states? Imaging data in PPPD appear to show both functional and structural changes compared to healthy controls, but there are some important confounding factors. Below is a summary of the key imaging findings described in patients with PPPD to-date.

### 3.1. Functional Imaging Study Findings

Riccelli et al. [31] found increased brain activity in the right middle insula and anterior to the central insular sulcus in healthy controls, but not in PPPD patients, who instead demonstrated increased activity in the visual cortex which correlated with the dizziness handicap inventory (DHI) score. Patients with PPPD have reduced resting-state connectivity between the hippocampi, particularly prominent for the left hippocampus, with reduced connectivity with the right inferior frontal gyrus, bilateral temporal lobes, bilateral insular cortices, bilateral central opercular cortices, left parietal opercular cortex, bilateral occipital lobes, and cerebellum (bilateral lobules VI and V, and left I–IV) [28]. Decreased connectivity in PPPD patients was also apparent between the temporal lobes, with the right amygdala and the right interior frontal gyrus and right orbitofrontal cortex, and between the cerebellum and the left caudate, left nucleus accumbens, and the precentral gyri (primary motor cortex) bilaterally. They concluded that reduced connectivity between visual and vestibular cortices, frontal regulatory regions and the hippocampi are the substrates for impaired spatial orientation and self-motion perception in PPPD. Reduced spontaneous activity has also been found in the right precuneus and cuneus in patients compared with controls [32]. In a subsequent study, the same authors demonstrated reduced connectivity within a network centred on the posterior cingulate gyrus and precuneus and implicated in self-centred cognition and attention monitoring [33].

Na et al. [34] found reduced perfusion of the left posterior insular cortex [28,35,36,37], and also in frontal lobe regions including the right inferior, bilateral superior frontal, and left medial orbital cortices in patients compared with controls. In contrast, PPPD patients had relative hyperperfusion of the cerebellum bilaterally, possibly related to increased computational demands on the cerebellum from increased visual attention and for postural control. Imaging findings likely reflect divergent functions and roles of the visual system in PPPD. Meanwhile, during a visual-motion paradigm, increased activity in the occipital cortex with increased connectivity to the auditory and somatosensory cortices in patients with PPPD may reflect re-weighting of multi-sensory inputs away from dysfunctional vestibular inputs to disambiguate between self and external motion [38,39].

### 3.2. Structural Imaging Study Findings

Structural brain changes have also been observed in patients with PPPD compared to healthy controls. Wurthmann et al. [40] found reduced grey matter volume in the following regions in PPPD patients compared to controls: (i) the left superior temporal gyrus; (ii) the bilateral middle temporal gyrus; (iii) the left motion-sensitive middle temporal visual area (MT/V5, implicated in visual motion perception [41]); (iv) the bilateral cerebellum; (v) the left-sided posterior hippocampus; (vi) the right precentral gyrus; (vii) the left anterior cingulate gyrus (involved in vestibular information processing, with increased cerebral glucose metabolism in acute vestibular impairment [42] and interoceptive awareness [43]; (viii) the left caudate nucleus (involved in vestibular and sensorimotor integration [44]); and (ix) the left dorsolateral prefrontal cortex. Nigro et al. [45] found reduced local gyrification in the superior and middle temporal gyri, supramarginal gyrus and right lateral occipital gyrus in patients with PPPD, with increased DHI score associated with reduced local gyrification in the right superior parietal lobule.

### 3.3. Limitations of Imaging Studies

Whilst imaging undoubtedly provides a relevant modality to explore neurological disease, current neuroimaging studies fall short of the optimal experimental design to understand the pathophysiology of PPPD (Table 1). Some pitfalls include the lack of a disease control group, with comparisons mainly being made to a healthy population. This means that it is not possible to comment on whether imaging changes are specific to PPPD, or whether they represent changes related to co-morbidities allied to the disorder (e.g., dizziness, anxiety, neuroticism, or postural instability). Secondly, confounding factors have not been systematically accounted for in most imaging studies, meaning that any brain changes cannot be exclusively attributed to PPPD. Finally, PPPD is a perceptual disorder with important behavioural correlates, and imaging studies have tended not to integrate perceptual or behavioural data with neuroimaging findings, hence failing to provide new information on the pathophysiology of PPPD. Additionally, the reviewed imaging studies concluded that there are no consistent and reproducible findings to explain the pathophysiological mechanisms in PPPD, but highlight the importance of emotional processing, which may have a bidirectional relationship with the pathophysiology of PPPD. A model that combines sensory sensitivity, self-reported low vision, and visual discomfort can explain up to 50% of PPPD symptom variance [2], supporting the theory that PPPD is a complex neurological disorder of multisensory processing, with a complex interaction between brain function and structure, emotion, perception, and behaviour, as recently proposed by Arshad and colleagues [11].

**Table 1 brainsci-12-00753-t001:** Summary of the data extraction from *n* = 8 articles focusing on pathophysiology of persistent postural-perceptual dizziness (PPPD) using brain imaging methods.

Author	Year	Country	Sample Size (*n*)	Age (y)	Active Control Group	Non-Imaging Outcome Data	Confounders Considered
				(Mean +/− SD)			
Wurthmann et al. [40]	2017	Germany	42 PPPD 42 Controls	PPPD—39.28 +/−7.55 Controls—37.97 +/− 9.93	No	DHI Impact of dizziness	Disease duration Active peripheral disease
Riccelli et al. [31]	2017	Italy, UK, USA	15 PPPD 15 Controls	PPPD—33.4 +/− 12.45 Controls—30.13 +/− 5.67	No	Visual motion stimuli DHI	Personality Motion sickness Anxiety/depression Active peripheral disease
Passamonti et al. [46]	2018	UK, Italy, USA	15 PPPD 15 Controls	PPPD—33.4 +/− 12.45 Controls—30.13 +/− 5.67	No	Visual motion stimuli	Neuroticism Motion sickness Anxiety/depression Active peripheral disease
Lee et al. [28]	2018	Republic of Korea, USA	38 PPPD 38 Controls	PPPD—48.6 +/− 12 Controls—47.5 +/− 13	No	DHI	Personality Motion sickness Anxiety Active neuro-otological disorder
Li et al. [33]	2020	China	10 PPPD 10 Controls	PPPD—47.70 +/− 12.37 Control—N/A	No	Disability DHI	Anxiety Depression Active peripheral disease
Na et al. [34]	2019	China	25 PPPD 25 Controls	PPPD—61.0 +/− 18.9 Controls—56.0 +/− 14.0	No	DHI	Duration of disorder Active vestibular disorder
Nigro et al. [45]	2019	Italy, USA, UK	15 PPPD 15 Controls	PPPD—33.4 +/− 12.4 Controls—30.1 +/− 5.6	No	DHI	Psychiatric comorbidities Anxiety/depression
Li et al. [32]	2020	China	12 PPPD 12 Controls	PPPD—44.25 +/− 10.73 Controls—N/A	No	Disability DHI	Anxiety Depression Active peripheral disease

## 4. Perception and Behaviour

Whilst it is tempting to focus on brain structure and function, imaging studies provide a static view of structure, connectivity, and activity that allow only a correlation with symptoms but cannot inform a causal link. The clinical features of PPPD highlight its essence as a primary perceptual disorder. That a functional disorder arises because of structural brain changes, and alteration in gyrification (cortical folding) is, in our view, difficult to account for in the context of current models of PPPD. However, perceptual dysfunction does of course lead to behavioural changes in posture and gait that may in turn effect alterations in cortical function and structure, the latter being a consequence, not a cause, of the perceptual disorder.

Behavioural data, therefore, seem crucial to propel our understanding of PPPD; understand the clinical symptoms, maladaptive strategies; and eventually facilitate the design and implementation of targeted effective therapies, most likely still involving physical therapy as a core domain. The current treatment options for PPPD include: vestibular rehabilitation (VR) and physiotherapy (PT); Cognitive Behavioural Therapy (CBT); medication; and electrical stimulation. VR, PT and CBT are the most common, either in isolation or combined; however, it is recommended that CBT is always implemented in addition to any physical therapy and vice versa [47], a framework that we have termed cognitive physical therapy [48].

Reports of postural control in patients diagnosed with chronic subjective dizziness [49] showed poorer performance in the Sensory Organization Test (SOT; a quantitative posturography assessment designed to assess an individual’s ability to utilise visual, vestibular, and proprioceptive cues for postural stability) than healthy controls and patients who had recovered from structural or metabolic illnesses that caused acute vertigo or dizziness [50]. Other posturography studies have revealed greater anteroposterior sway associated with visual dependence and greater mediolateral sway in low demand conditions than controls, confirming the notion that PPPD induces physical changes in posture. SOT performance did not discriminate patients with PPPD from those that had recovered from an acute vestibulopathy with enough sensitivity and specificity to serve as a diagnostic biomarker. However, visual preference patterns, poorer performance on more challenging postural conditions [51], and higher variability across the six conditions identified patients with PPPD with high specificity (>94%), but poor sensitivity (≤55%). Automated postural and gait parameters in other studies showed promise for differentiating patients with peripheral vestibular, cerebellar, and functional balance disorders [52,53,54]. These results suggest that stance and gait metrics could become potential biomarkers of PPPD, but as with imaging studies, many posturography studies compare PPPD patients against controls, where segregating abnormalities from patients is far less challenging than differentiating different patient populations.

Particular postural metrics have been identified in patients with functional balance disorders, such as a stiffened ankle strategy with a high frequency (3.5–8 Hz) of body sway in patients with phobic postural vertigo [13]—a previous term for PPPD related to the tendency to abnormally recruit afferent feedback loops during quiet stance [15]. Such patients use postural control strategies appropriate for situations that may threaten postural stability that become pervasive and contextually inappropriate [29]. One consequence of postural threat is the reduced threshold for detecting visual targets [55], so perceived threat may promote the upweighting of visual cues, in line with visual motion discomfort in these patients (perception) and greater sway when faced with concurrent visual motion stimuli (behaviour). Threat perception influences objective and subjective postural control even in healthy individuals when primed with information regarding an upcoming postural task [56]. It is unsurprising, then, that threat could modify postural control in patients with PPPD. Indeed, in patients with functional gait disorders, a tap to the shoulders, without inducing any postural instability, is sufficient to generate destabilizing postural reactions, presumably due to abnormal anticipatory behaviour [57]. Such anticipation of a potential postural threat may have diagnostic and treatment applications for conditions related to PPPD.

We have previously highlighted the possible role of abnormal magnitude estimation in the generation of functional balance disorders [11], linked perhaps to reduced right-lateralised vestibulo-cortical dominance, which leads to distorted representations of space and time, due to biased perceptions towards larger magnitudes.

Indeed, at height, normal individuals overestimate magnitudes of forward leaning by up to 10% and increased soleus responses to Achilles’ tendon taps by almost 20% relative to being at ground level [58,59], suggesting that humans adjust the gain of afferent sensory signals proportionate to the perceived postural threat. In line with these observations, patients with PPPD tend to over-estimate head roll tilt substantially more than patients with unilateral vestibular hypofunction [60].

These findings suggest that patients with PPPD rely on threat-related postural control strategies that deviate from contextually optimal control, such as a stiffened ankle-strategy, a lower threshold for engaging control loops that incorporate visually dependent feedback, increased sensitivity to motion, and increased gain of posture-related motor reflexes [29,61]. Neuroimaging studies showing decreased activity and connectivity of vestibular cortical regions to vestibular stimulation (i.e., vestibular downweighing [28,31,32,33,34]), but increased activity and connectivity of frontal–occipital networks to visual motion stimulation (i.e., visual upweighting), thus align with these behavioural data.

## 5. Integrating Brain Changes with Behaviour and Perception

Accordingly, to study brain images in isolation and not to correlate the data with clinical features of PPPD may not lead us to decipher the pathophysiology of PPPD—a perceptual disorder with behavioural consequences. The core symptoms of PPPD seem to arise from multi-level changes in brain function, perception, and behaviour, with a diminished sense of agency about the control of postural control. We have previously proposed that the pathophysiological mechanisms underlying PPPD involve changes in processing of multi-modal stimuli related to our immediate environment and our movement within this. Such information requires ascending brainstem pathways to be appropriately processed at a cortical level. Any alteration in conscious awareness of (self and external) motion can induce a reduced tolerance to perceived postural instability, generating a perception of heightened instability, and thus resulting in adjustments to locomotor control strategies [11]. Furthermore, bottom-up (afferent sensory signals to cortical structures) and top-down (cortical to motor effectors) alterations in functioning are affected by hypervigilant states, such as elevated trait and postural anxiety. These in turn induce greater behavioural and perceptual responses to any discrepancies between actual sway and perceived self-motion, namely a stiffening of posture and increased subjective instability that drive further hypervigilant states. Perceptually biased estimates of motion magnitude feed into an erroneous prediction of self-motion perception [11]. Critically, these abnormalities would be expected to correlate with psychological measures such as trait anxiety, neuroticism, and negative illness behaviour, although such variables are likely specific to postural control and may not be faithfully captured by standard psychological questionnaires [11]. Anecdotally, we have observed that many patients with PPPD deny any perception of anxiety on a day-to-day basis, but often display features of postural anxiety during behavioural postural tasks (e.g., Romberg stance with eyes closed).

Considering the postural nature of patients’ complaints, understanding the pathophysiology of PPPD is likely to require dynamic behavioural data, perhaps not *exclusively* postural. Previously, Furman and colleagues [62] have shown a significantly higher vestibulo-ocular reflex (VOR) gain and a shorter VOR time constant in patients with anxiety compared to healthy controls. This led them to hypothesise that in patients with anxiety, there is increased vestibular sensitivity and impaired velocity storage, and that functional dizziness may alter vestibular sensitivity via a downweighting of central vestibular pathways. Schroder and colleagues [63] used an interesting oculomotor paradigm in eight patients with PPPD, whereby subjects performed combined head–eye gaze shifts to large eccentric targets, with and without experimentally weighted head characteristics. They measured the amount of compensatory eye movements for gaze stabilisation during time windows that either engage motor planning and internal expectations or do not. They found that sensory-driven gaze stabilisation was intact, whereas internally driven gaze stabilisation was markedly altered in functional dizziness, consistent with the notion that internal expectations are mismatched to actual body states [63]. Such studies support the need for behavioural data for understanding the pathophysiology of a perceptual disorder where “standard” clinical assessments are normal.

Task-based functional imaging may help uncover the basis for visual hypersensitivity, which frequently accompanies PPPD, but is less likely to inform the mechanism of perceived postural instability, the core feature of this disorder. Ultimately, PPPD research will benefit from a combination of methodologies that include postural assessments, perceptual psychophysical measures, psychological measures, and how these interact with cortical and subcortical dysfunction across brain areas (imaging) and networks mediating such functions. The latter could be achieved through a range of non-invasive techniques such as electroencephalography [64], magnetoencephalography [65], or functional near-infrared spectroscopy [66].

## 6. Conclusions

There has been some great progress in our understanding of PPPD, and a desire to improve treatments for one of the most common dizziness presentations in Neurology. Clinical studies would be strengthened, however, by experimental work exploring whether transient disruption of critical brain networks—using non-invasive brain stimulation, for example—could reproduce similar symptoms to those seen in PPPD in otherwise healthy controls. Moreover, understanding the role of altered magnitude estimates may help explain the development and perpetuation of symptoms, and the role of reduced vestibulo-cortical dominance in the generation of distorted representations of the magnitude-space-time continuum. Work is ongoing to understand the predictors of PPPD in order to intervene at the earliest possible stage and avoid development of chronic dizziness symptoms. In patients with established PPPD, however, the development of biomarkers to track progress and measure response to therapies needs to incorporate abnormal perception, not just posturography deficits or imaging abnormalities, given the degree of heterogeneity in behavioural adaptations to a perceived postural threat.

## Figures and Tables

**Figure 1 brainsci-12-00753-f001:**
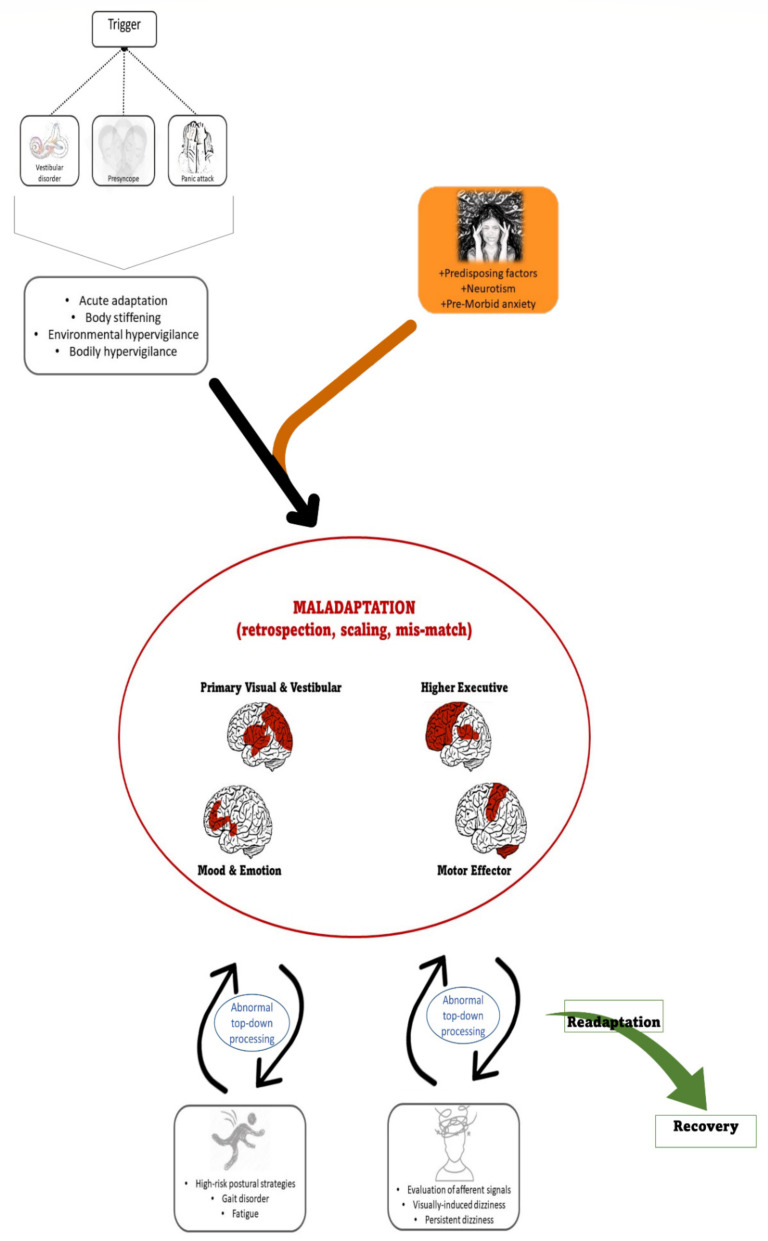
Pathophysiology of persistent postural-perceptual dizziness and neural networks involved. A triggering event may lead to an acute adaptation phase with transient behavioural and perceptual changes. In the context of predisposing psychological factors, such an adaptation becomes a maladaptive strategy, involving abnormal interactions between primary visual and vestibular cortices, higher executive cortical areas, limbic structures that process mood and emotion, and motor efferent areas. Maladaptation induces heightened introspection and abnormal interpretation of afferent sensory signals (sensory scaling mis-match), which drives typical symptoms of PPPD, including visually induced dizziness, persistent dizziness, gait disorder, and cognitive fatigue.

**Figure 2 brainsci-12-00753-f002:**
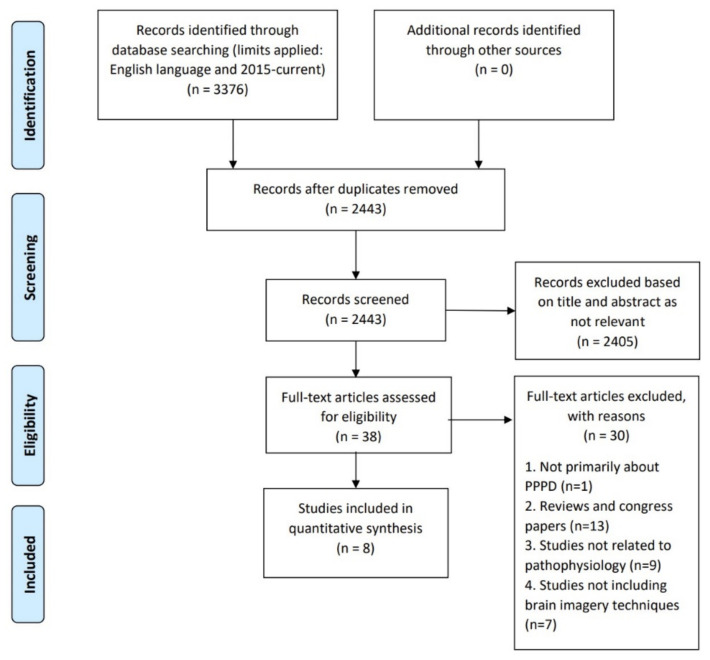
Flowchart depicting the paper selection procedure for the systematic review of neuroimaging data in PPPD patients (based on PRISMA 2009 Flow Diagram guidelines).

## Data Availability

Not applicable.

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
