# Peer review of "Persistent Postural-Perceptual Dizziness (PPPD) from Brain Imaging to Behaviour and Perception"

_brainsci, 2022, doi:10.3390/brainsci12060753_

Round 1

Reviewer 1 Report

This is a very interesting Perspective paper about the integration of neuroimaging studies and behavioural measurements in PPPD (persistent postural-perceptual dizziness). PPPD is a relatively new concept of dizziness with diagnostic criteria defined in 2017 by a consensus committee of the Barany society.

The authors argue to explore the relationship between behavioural postural changes, perceptual abnormalities, and imaging correlates in PPPD. Behaviour, perception, and cortical and subcortical brain function as well should be analyzed to integratively explore the pathophysiology of PPPD.

This is a well conceived paper which is very interesting to read. The argumentation is conclusive and the perspective paper may be able to inspire the research concepts in this field.

I only have few minor suggestions:

As the concept of PPPD is relatively new it should be mentioned that the diagnostic criteria of PPPD were defined in 2017 by a consensus committee of the Barany society. The reference is already cited in the manuscript.

As not every reader of the journal will be familiar with the concept of PPPD it may be easier to understand if the major characteristics and factors playing a role in PPPD were summarized in a figure.

In line 51 the authors state a mismatch between expected and actual motion signals. That reminds me of the concept of predictive coding. See for instance Philos Trans R Soc Lond B Biol Sci. 2005 Apr 29;360(1456):815-36. doi: 10.1098/rstb.2005.1622.  or  Neurosci Biobehav Rev. 2016 Dec;71:379-387. doi: 10.1016/j.neubiorev.2016.09.009.

Author Response

We are very grateful to the Editors and Reviewers for the feedback, comments, and suggestions. Please find below a point by point response to each of the Reviewers’ comments/ questions/ suggestions in bold, the responses in black, and the actual changes made in the revised manuscript in red. We believe the changes have strengthened the manuscript and are again grateful to the Reviewers.

Reviewer 1

This is a very interesting Perspective paper about the integration of neuroimaging studies and behavioural measurements in PPPD (persistent postural-perceptual dizziness). PPPD is a relatively new concept of dizziness with diagnostic criteria defined in 2017 by a consensus committee of the Barany society.

The authors argue to explore the relationship between behavioural postural changes, perceptual abnormalities, and imaging correlates in PPPD. Behaviour, perception, and cortical and subcortical brain function as well should be analyzed to integratively explore the pathophysiology of PPPD.

This is a well-conceived paper which is very interesting to read. The argumentation is conclusive and the perspective paper may be able to inspire the research concepts in this field.

I only have few minor suggestions:

Q1: As the concept of PPPD is relatively new it should be mentioned that the diagnostic criteria of PPPD were defined in 2017 by a consensus committee of the Barany society. The reference is already cited in the manuscript.

R1: We are extremely grateful to the Reviewer for the pertinent and very helpful comments. We have added a sentence stating that the diagnostic criteria for PPPD were defined in 2017, citing the reference.

“In 2017, a consensus committee of the Bárány society proposed diagnostic criteria for PPPD that included”

Q2: As not every reader of the journal will be familiar with the concept of PPPD it may be easier to understand if the major characteristics and factors playing a role in PPPD were summarized in a figure.

R2: We have drafted a Figure summarising the main features and pathophysiological concepts in PPPD.

“Figure 1: Pathophysiology of Persistent Postural-Perceptual Dizziness and neural networks involved. A triggering event may lead to an acute adaptation phase with transient behavioural and perceptual changes. In the context of predisposing psychological factors, such adaptation becomes a maladaptive strategy, involving abnormal interactions between primary visual and vestibular cortices, higher executive cortical areas, limbic structures that process mood and emotion, and motor efferent areas. Maladaptation induces heightened introspection, and abnormal interpretation of afferent sensory signals (sensory scaling mis-match) that drives typical symptoms of PPPD, including visually-induced dizziness, persistent dizziness, gait disorder, and cognitive fatigue.”

Q3: In line 51 the authors state a mismatch between expected and actual motion signals. That reminds me of the concept of predictive coding. See for instance Philos Trans R Soc Lond B Biol Sci. 2005 Apr 29;360(1456):815-36. doi: 10.1098/rstb.2005.1622.  or  Neurosci Biobehav Rev. 2016 Dec;71:379-387. doi: 10.1016/j.neubiorev.2016.09.009.

R3: We agree that predictive coding is indeed relevant to current concepts of PPD and have added the following sentence to support this:

“Indeed, it has been suggested that the brain uses generative models to actively construct explanations to infer the causes of sensory inputs, such that the brain’s internal model of the external world is optimized in such a way that sensory inputs are predictable (17, 18). Prediction error is therefore the difference between the input observed and that predicted by the generative model and inferred causes. The concept of predictive coding describes the brain’s attempt to minimise surprise and resolve uncertainty about sensory information - reducing the mismatch between unpredicted (‘salient’) sensations and those predicted under the generative model (17). Only unpredicted inputs must then be further transmitted and analysed. The abnormal processing of seemingly ‘unpredicted’ signals may account for abnormal prediction error and increased perception of sway in patients with PPPD. Within this framework of predictive coding, central processing of incoming sensory information is biased by a mismatch resulting from incorrect internal expectations leading to the perception of impaired posture or balance.”

Reviewer 2 Report

The authors present a deeply researched and comprehensive review on the pathophysiology of PPPD, how imaging studies have mostly failed to contribute to our understanding thereof, and how the field could move forward. It is a timely and well written contribution to the field.

Regarding the summary of imaging studies the authors should make sure that the list of studies is complete (10.3389/fnbeh.2015.00334 is missing, maybe others). To that end, they may consider adding a brief note on methodology (i.e. search strategy) or maybe a table with information on the studies that can convey the overall quality of available studies, eg. data on sample size (power analysis for sample size estimation provided?), available control group, assessment of outcomes (DHI, task performance etc), assessment of confounders (eg anxiety assessment in control groups), imaging method.

In their discussion of the limitations of imaging studies the authors concede that the imaging studies to date “fall short of optimal experimental design”. This is a lenient sentence and is soon countered by the conclusion that studies provide support for disruption of multidensory integration. I am not conviced this is a well-supported conclusion. What the studies to date show us (in my opinion) is that there is no simple and reproducible finding that helps us understand PPPD, and that without clinical controls, assessment of confounders and without integrating perceptual/behavioral tasks neuroimaging will not shed any new light on the pathophysiology of PPPD beyond “it’s complicated”.

Towards the end of the paper the authors list EEG, MEG and infrared spectroscopy as promising assessments. It would be interesting to read whether task-based fMRI holds any promises to understanding PPPD better, or whether the authors think that testing PPPD in a supine position will never really bring us forward.

When discussing the influence of threat perception, the authors could mention a recent study which indicates that cognitive priors regarding postural threat influence postural control in healthy participants. (10.1016/j.gaitpost.2022.02.015)

When discussing how behavioural data should guide treatment developmen (p. 4, lines 164-167) the authors could cite a recent summary of current PPPD treatment strategies (10.1007/s11940-018-0535-0).

The shoulder tap test, which the senior author co-published, should be noted as an excellent clinical diagnostic application of the abnormal responses to perceived postural threat in patients with functional disorder of gait and stance. 10.1212/WNL.0000000000012886 (eg on page 5, Line 237).

In the following segment citations to respective imaging studies should be added for clarity: „Neuroimaging studies showing decreased activity and connectivity of vestibular cortical regions to vestibular stimulation (i.e., vestibular downweighing), but increased activity and connectivity of frontal-occipital networks to visual motion stimulation (i.e., visual upweighting), thus align with these behavioural data.“

Lastly, a short paragraph on oculomotor aspects of PPPD pathophysiology would be interesting. Older work by RG Jacob or a recent study by a Munich group (10.3389/fnins.2021.685590) could be relevant.

Author Response

We are very grateful to the Editors and Reviewers for the feedback, comments, and suggestions. Please find below a point by point response to each of the Reviewers’ comments/ questions/ suggestions in bold, the responses in black, and the actual changes made in the revised manuscript in red. We believe the changes have strengthened the manuscript and are again grateful to the Reviewers.

Reviewer 2

The authors present a deeply researched and comprehensive review on the pathophysiology of PPPD, how imaging studies have mostly failed to contribute to our understanding thereof, and how the field could move forward. It is a timely and well written contribution to the field.

Q1: Regarding the summary of imaging studies the authors should make sure that the list of studies is complete (10.3389/fnbeh.2015.00334 is missing, maybe others). To that end, they may consider adding a brief note on methodology (i.e. search strategy) or maybe a table with information on the studies that can convey the overall quality of available studies, eg. data on sample size (power analysis for sample size estimation provided?), available control group, assessment of outcomes (DHI, task performance etc), assessment of confounders (eg anxiety assessment in control groups), imaging method.

R1: We are grateful for this comment. We acknowledge that for the reader it would indeed be helpful to briefly summarise this information (particularly highlighting some of the limitations of the available imaging literature on PPPD). We have therefore provided a summary table as suggested and have added the following paragraph. Note that this review has focused strictly on studies that included PPPD patients to avoid diagnostic bias from previous studies conducted prior to the introduction of the diagnostic criteria.

“A systematic literature review was completed to identify English-language, original research articles using neuroimaging to investigate patients with PPPD as defined in the ICD-11 criteria (ICD-11, 2005) published between 1st January 2015 to 30th September 2020. Information sources used in this review were Medline/Ovid, Embase/Ovid and PubMed. The search strings included the following key terms “Vestibular Diseases” OR “Dizziness” OR “persistent postural perceptual dizziness”; OR “persistent perceptual postural dizziness”; OR (“persistent postural”; and “perceptual dizziness”) AND pathophysiology OR physiopathology OR pathophysiological mechanism; OR pathogenesis OR pathological OR “psychophysics”. The search criteria returned 3376 articles, reduced to 2443 after removal of duplicates. After screening the title and abstract a further 2405 were excluded as irrelevant. The remaining 38 full-text articles were assessed for eligibility and 30 excluded due to not exclusively relating to PPPD (n=1), review articles (n=13), studies not related to pathophysiology of PPPD (n=9) and studies not including neuroimaging (n=7) (figure 2). The remaining 8 articles were evaluated for the risk of bias, analysed and the appropriate data was extracted. For a comprehensive imaging review see also (30).”

“Figure 2: Flowchart depicting the paper selection procedure for the systematic review of neuroimaging data in PPPD patients (based on PRISMA 2009 Flow Diagram guidelines)”

“Table 1: Summary of the data extraction from n=8 articles focusing on pathophysiology of persistent postural-perceptual dizziness (PPPD) using brain imaging methods.”

Q2: In their discussion of the limitations of imaging studies the authors concede that the imaging studies to date “fall short of optimal experimental design”. This is a lenient sentence and is soon countered by the conclusion that studies provide support for disruption of multidensory integration. I am not conviced this is a well-supported conclusion. What the studies to date show us (in my opinion) is that there is no simple and reproducible finding that helps us understand PPPD, and that without clinical controls, assessment of confounders and without integrating perceptual/behavioral tasks neuroimaging will not shed any new light on the pathophysiology of PPPD beyond “it’s complicated”.

R2: We agree that we have been rather ‘soft’ in our critique and have amended the sentence thus:

“Some pitfalls include the lack of a disease control group, with comparisons in the main being made to a healthy population.  This means that it is not possible to comment on whether imaging changes are specific to PPPD, or whether they represent changes related to co-morbidities allied to the disorder (e.g., dizziness, anxiety, neuroticism, or postural instability). Secondly, confounding factors have not been systematically accounted for in most imaging studies, meaning that any brain changes cannot be exclusively attributed to PPPD. Finally, PPPD is a perceptual disorder with important behavioral correlates, and imaging studies have tended not to integrate perceptual or behavioural data with neuroimaging findings, hence failing to provide new information on the pathophysiology of PPPD. Additionally, the reviewed imaging studies have concluded that there are no consistent and reproducible findings to explain the pathophysiological mechanisms in PPPD, yet highlighting the importance of emotional processing, which may have a bi-directional relationship to the pathophysiology of PPPD.  A model that combines sensory sensitivity, self-reported low vision and visual discomfort can explain up to 50% of PPPD symptom variance (2), supporting the theory that PPPD is a complex neurological dis-order of multisensory processing, with a complex interaction between brain function & structure, emotion, perception, and behaviour, as recently proposed by Arshad and colleagues (11).”

Q3: Towards the end of the paper the authors list EEG, MEG and infrared spectroscopy as promising assessments. It would be interesting to read whether task-based fMRI holds any promises to understanding PPPD better, or whether the authors think that testing PPPD in a supine position will never really bring us forward.

R3: The reviewer raises an important and interesting point. We have speculated on this in the revised version, thus:

“Considering the nature of the postural nature of patient’s complains, understanding the pathophysiology of PPPD is likely to require dynamic behavioural data. Task-based functional imaging may help uncover the basis for visual hypersensitivity, that frequently accompanies PPPD, but is less likely to inform the mechanism of perceived postural instability, the core feature of this disorder. Ultimately, PPPD research will benefit from a combination of methodologies that include postural assessments, perceptual psychophysical measures, psychological measures, and how these interact with cortical and subcortical dysfunction across brain areas (imaging) and networks mediating such functions. The latter could be achieved through a range of non-invasive techniques such as electroencephalography 54, magnetoencephalography 55, or functional near infrared spectroscopy 56.”

Q4: When discussing the influence of threat perception, the authors could mention a recent study which indicates that cognitive priors regarding postural threat influence postural control in healthy participants. (10.1016/j.gaitpost.2022.02.015).

R4: We are grateful for this comment. We have added a statement to this effect, including the suggested reference.

“Threat perception influences objective and subjective postural control even in healthy individuals when primed with information regarding an upcoming postural task (55). It is unsurprising then, that threat could modify postural control in patients with PPPD.”

Q5: When discussing how behavioural data should guide treatment developmen (p. 4, lines 164-167) the authors could cite a recent summary of current PPPD treatment strategies (10.1007/s11940-018-0535-0).

R5: We have added the reference suggested by the Reviewer regarding treatment development strategies:

“Current treatment options for PPPD include vestibular rehabilitation (VR) and physiotherapy (PT); Cognitive Behavioural Therapy (CBT); medication and electrical stimulation. VR, PT and CBT are the most common, either in isolation or combined, however it is recommended that CBT is always implemented in addition to any physical therapy and viceversa (46), a framework that we have termed cognitive physical therapy (47).”

Q6: The shoulder tap test, which the senior author co-published, should be noted as an excellent clinical diagnostic application of the abnormal responses to perceived postural threat in patients with functional disorder of gait and stance. 10.1212/WNL.0000000000012886 (eg on page 5, Line 237).

R6: We are grateful for the reviewer’s comment. We have added the following statement, and reference, as suggested.

“Indeed, in patients with functional gait disorders, a tap to the shoulders, without inducing any postural instability, is sufficient to generate destabilizing postural reactions, presumably due to abnormal anticipatory behaviour (56). Such anticipation to a potential postural threat may have diagnostic and treatment applications for related conditions such as PPPD.”

Q7: In the following segment citations to respective imaging studies should be added for clarity: „Neuroimaging studies showing decreased activity and connectivity of vestibular cortical regions to vestibular stimulation (i.e., vestibular downweighing), but increased activity and connectivity of frontal-occipital networks to visual motion stimulation (i.e., visual upweighting), thus align with these behavioural data.“

R7: We apologise for this oversight. We have added the relevant references in this section.

“Neuroimaging studies showing decreased activity and connectivity of vestibular cortical regions to vestibular stimulation (i.e., vestibular downweighing) (28, 31-34), but increased activity and connectivity of frontal-occipital networks to visual motion stimulation (i.e., visual upweighting), thus align with these behavioural data.”

Q8: Lastly, a short paragraph on oculomotor aspects of PPPD pathophysiology would be interesting. Older work by RG Jacob or a recent study by a Munich group (10.3389/fnins.2021.685590) could be relevant.

R8: We are grateful for this comment and for directing us towards this interesting research area. We have added a short section regarding oculomotor function in functional dizziness:

“Considering the nature of the postural nature of patient’s complains, understanding the pathophysiology of PPPD is likely to require dynamic behavioural data, perhaps not exclusively postural. Previously, Furman and colleagues (61) have shown a significantly higher vestibulo-ocular reflex (VOR) gain and a shorter VOR time constant in patients with anxiety compared to healthy controls. This led them to hypothesise that in patients with anxiety, there is increased vestibular sensitivity and impaired velocity storage, and that functional dizziness may alter vestibular sensitivity via a down-weighting of central vestibular pathways. Schroder and colleagues (62) used an interesting oculomotor paradigm in eight patients with PPPD, whereby subjects performed combined head-eye gaze shifts to large eccentric targets, with and without experimentally weighted head characteristics. They measured the amount of compensatory eye movements for gaze stabilization during time windows that either engage motor planning and internal expectations or do not. They found that sensory-driven gaze stabilization was intact, whereas internally-driven gaze stabilization was markedly altered in functional dizziness, consistent with the notion that internal expectations are mis-matched to actual body states [REF]. Such studies support the need for behavioural data in understanding the pathophysiology of a perceptual disorder where ‘standard’ clinical assessments are normal.”